# RNNs Based Planning with Discrete and Continuous Actions

## Abstract

Dealing with discrete and continuous changes in real-world dynamic environments is of great importance for robots. Despite the success of previous approaches, they impose severe restrictions, such as convex quadratic constraints on state variables, which limits the expressivity of the problem, especially when the problem is non-convex. In this paper, we propose a novel algorithm framework based on recurrent neural networks. We cast the mixed planning with discrete and continuous actions in non-convex domains as a gradient descent search problem. In the experiment, we exhibit that our algorithm framework is both effective and efficient, especially when solving non-convex planning problems.

## Introduction

To control robots with many degrees of freedom and very complicated dynamics, robotics community has made tremendous success with trajectory optimization and other sophisticated methods. The robotics community, however, often neglects activity planning and resorts to chaining complex behaviors computed with trajectory optimization manually. Those approaches were restricted to limited horizons in which fixed time discretization works well. They cannot handle the problem over long horizons where activity planning is required. Reasoning with both discrete and continuous behaviors is necessary for better performance. *For example, in an ocean mission described below, a ship navigates with an ROV including discrete actions, such as deploying an ROV, and continuous actions that involve the continuous dynamic transition of the ship. Trajectory planning cannot handle missions mixed with discrete actions and continuous actions.*

Heuristic forward search approaches for planning have shown immense progress. For example, Metric-FF (Hoffmann 2003) handles a mix of discrete and continuous effects and preconditions by ignoring all effects that decrease the value of affected variable. However, it is limited to fixed continuous variables and simple numerical effects. *As an example, we demonstrate a valid plan in* ocean mission scenario *in Figure 1, where the blue path is navigated by the ship, the orange path is navigated by ROV and the green path*

*is navigated by AUV. The squares denoted by A, B and C are places to be visited and the black regions are obstacles.* To handle missions with a mix of discrete and continuous actions, many approaches have been proposed such as COLIN (Coles et al. 2012). However, COLIN only supports continuous time-dependent effects with constant rates of change and linear state variable constraints, which makes it cannot scale well to practical problems.

A complex realistic planning mission is combined with trajectory planning and classical planning. In order to generate plans for realistic planning missions, many excellent approaches have been proposed. Kongming (Li and Williams 2008) was one of the first approaches to generate a practical plan for real robots with merged discrete classical planning and continuous trajectory optimization. However, Kongming is based on a fixed time discretization and it can not handle missions with short and long-time activities coexist. To improve on that, ScottyActivity (Fernández-González, Williams, and Karpas 2018) and ScottyPath (Fernández-González 2018) was proposed. ScottyActivity can handle a mix of discrete and continuous action in a convex domain. To be able to handle planning problems with obstacles, ScottyPath is based on ScottyActivity with constructing convex safe regions to avoid obstacles and informed search. Although safe regions generation can consider obstacles, there could be a shorter plan avoiding all obstacles and going through areas not covered by the safe regions. As shown in Figure 1(b), blue areas are safe regions divided by ScottyPath and each step can not include red strips areas, each movement should be involved in safe regions (blue areas). But the plan can be much shorter when crossing the red strips region $R_1$ directly.

Here is an ocean mission scenario example, a ship equips an AUV (autonomous underwater vehicle) and an ROV (remotely operated vehicle). The AUV needs to take images at region A, and the ROV needs to take samples in regions B and C. All three vehicles need to avoid obstacles (the black areas) and reach destination regions (the yellow area) at the end. Each movement needs three parameters, a x-velocity $v_x$, a y-velocity $v_y$ and a duration $t$. In particular, if the ship deploys the ROV, which can move within a circle centered at the ship with a radius $R$, the ship needs to remain at the deployment location until the ROV is recovered again. Figure

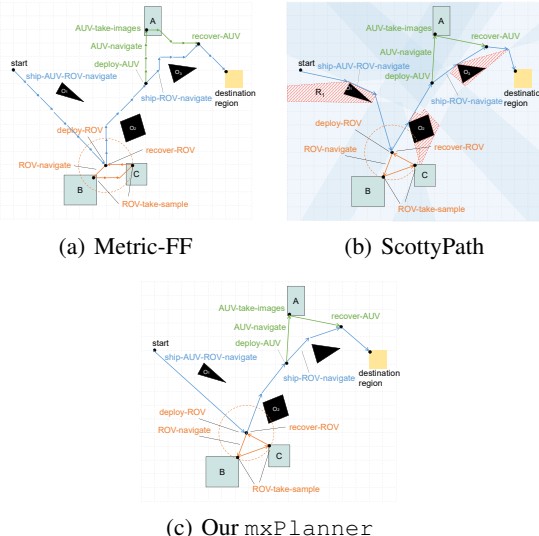

(a) Metric-FF

(b) ScottyPath

(c) Our `mxPlanner`

Figure 1: Valid plans based on Metric-FF, ScottyPath and our approach in ocean mission scenario, blue areas in (b) are safe regions, red strips areas are not safe.

1(a) shows a valid plan based on Metric-FF. In this example, we assign all three parameters to values of 1 to owing to fixed continuous effects of Metric-FF. Another plan based on ScottyPath as illustrated in Figure 1(b). ScottyPath separates the mission area into the safe regions (blue areas) and the others (red strips areas). Each step must be in a same safe region. However there may be a shorter valid plan not covered by the safe regions just shown as Figure 1(c). Our algorithm shows three advantages, first, unlike Metric-FF, `mxPlanner` does not need human computation in advance. Second, comparing with ScottyActivity, `mxPlanner` can handle a domain with obstacles. Finally, `mxPlanner` does not divide a area into pieces avoiding searching space loss comparing with ScottyPath.

In this paper, we present an algorithm framework based on recurrent neural networks (RNNs) and heuristic searching to generate an obstacle-free plan with a mix of discrete and continuous actions. In the remainder of the paper, we first introduce related works and a formal definition of our problem. After that, we present our approach in detail and evaluate our approach by comparing it to previous approaches to exhibit the effectiveness. Finally, we conclude the paper with future work.

## Related Work

As for realistic planning missions, generating trajectories from an initial state to goals is a fundamental one in Automated Planning research. Many approaches have been used extensively to real-world robot planning missions with complex dynamics for purposes of time minimization, resource usage minimization or obstacles avoidance, and we just illustrate a few examples. Moon et al. (Moon and Prasad 2011) proposed a minimum time approach for obstacles avoidance by Non-Linear Trajectory Generation. A tree-based algo-

rithm (Langelaan 2008) is used to find a feasible minimum energy trajectory from a start position to a distant goal by precomputing a set of branches from the space of allowable inputs. And Gracia et al. (Gracia, Sala, and Garelli 2012) developed a supervisory loop to fulfill workspace constraints caused by robot mechanical limits, collision avoidance, and industrial security in robotic systems with geometric invariance and sliding mode related concepts. Evolutionary algorithms have also been proposed to solve trajectory planning problems (Li et al. 2016; Alvarez, Caiti, and Onken 2004; Nikolos et al. 2003; Zheng et al. 2005; Guo and Gao 2009; Fu, Ding, and Zhou 2011; Zhang et al. 2014; Liu, Yu, and Dai 2008). Despite the success of those approaches, they cannot solve trajectory planning problems involving both continuous and discrete actions.

There have been considerable advancements in the development of classical planning to solve state-dependent goals planning missions. Most of them succeed by discrete heuristics searching function or backward chaining search. FF (Hoffmann 2001) relies on forwarding search in the state space, guided by goal distances estimated by ignoring delete lists. A heuristic computation algorithm (Chatterjee et al. 2019) was proposed applying to real-world robotic missions using conservative edges in the heuristic-space for reducing state expansions without considering continuous effects. To handle continuous effects and preconditions, Metric-FF (Hoffmann 2003) was built based on discretization and ignoring all effects that decrease the value of the affected variable. Although COLIN (Coles et al. 2012) and OPTIC++ (Denenberg and Coles 2019) does not rely on discretization and they can handle missions with a mix of discrete and continuous actions, they can only support either continuous time-dependent effects with "constant" rates of change, or monotonic continuous change of variables. They cannot deal with non-convex problems, e.g., with obstacles in trajectory planning problems, as done by our `mxPlanner` approach.

Researchers have considered combining classical planning with trajectory planning to solve realistic planning missions mixed with discrete and continuous actions. Kongming (Li and Williams 2008) introduced a novel approach to solve planning problems with hybrid flow graph which is capable of representing continuous trajectories in a discrete planning framework. However, Kongming is not able to handle realistic planning missions for its fixed time discretization. POPCORN (Savas et al. 2016) introduced continuous control parameters in PDDL to allow infinite parameters in actions. However, it can only be used in discrete numeric effects. (Wu, Say, and Sanner 2017) proposed to handle non-linear continuous effects with gradient descent. However, they did not handle scenarios with a mix of discrete and continuous preconditions.

## Problem Definition

We define a planning problem as a tuple $\mathcal{M} = \langle \mathcal{S}, \mathcal{A}, s_0, g \rangle$. $\mathcal{S}$ is a set of states, each of which is composed of a set of propositions (e.g. $(equip\ (ship\ AUV))$), including numerical equations (e.g. $(=\ duration\ 0)$). For a state $s$, we define $\vec{s}$ is the value vector of all the numeric variables.

**(a) Action models**

```
deploy-ROV( )
pre: (equip(ship ROV))
eff: (not (equip (ship ROV)))
ROV-navigate(?v_x-R,?v_y-R,?t-R)
pre: (not (equip (ship ROV)))
eff: (increase location-x_ROV(* ?v_x  ?t))
     (increase location-y_ROV(* ?v_y  ?t))
     (increase total-time ?t̄ )
     ...
```

**(b) Goal**

```
goal:
(image-taken A)
(sample-taken B)
(sample-taken C)
(mission-completed)
```

**(c) Initial state**

```
initial state:
(equip (ship AUV))
(equip (ship ROV))
(= location-x_ship 1)
(= location-y_ship 11)
(= location-x_AUV 1)
(= location-y_AUV 11)
(= location-x_ROV 1)
(= location-y_ROV 11)
(= O_1  φ((4.2,9.3),(4.6,10),(6.4,8.7)))
(= O_2  φ((9,7.2),(10.5,7.8),(11,6.3),(9.5,5.5)))
(= O_3  φ((12.5,11.5),(15,11),(13,10)))
(= A   φ((11,13.5),(12,13.5),(12,15.5),(11,15.5)))
(= B   φ((5,1),(7.2,1),(7.2,3.1),(5,3.1)))
(= C   φ((9.5,3.5),(11,3.5),(11,5.2),(9.5,5.2)) )
(= destination-region  φ((17,9.8),(18.5,9.8),(18.5,11),(17,11)))
```

**(d) An output of the ocean mission**

```
plan:
"ship-navigate (3, 3.5, 2)"→"deploy-ROV()"→"ROV-navigate(-1, -2, 1)"→"ROV-take-sample(B)"→"ROV-navigate(1, 2, 1)"
→"ROV-take-sample(C)"→"ROV-navigate(-1, 1, 1)"→"recover-ROV"→"ship-navigate(1, 3, 1)"→"ship-navigate(1, 1, 2)"
→"deploy-AUV"→"AUV-navigate(0, 2, 2)"→"ship-navigate(1, 1.2, 1.5)"→"AUV-take-images(A)"
→"ship-navigate(3, 1, 0.8)"→"AUV-navigate(4, -1, 1)"→"recover-AUV"→"ship-navigate(1, -1, 2)"
```

Figure 2: An Ocean mission example, where (a), (b), (c) are input of our problem, (d) is an output of the mission.

$\mathcal{A}$ is a set of action models, including discrete and continuous action models. Each action model is a tuple $\langle a, pre(a), eff(a)\rangle$, where $a$ is an action name with zero or more parameters, $pre(a)$ is a set of preconditions that should be satisfied when $a$ is executed, and $eff(a)$ is a set of effects that update the state where $a$ is executed. Considering preconditions of $a$ can be either numeric or propositional, we denote $pre(a)$ by $pre(a) = pre_\leftrightarrow(a) \cup pre_\perp(a)$, where $pre_\leftrightarrow(a)$ is a set of numeric preconditions, e.g., "$(\geq ?x\ ?lower)$" indicating the x-axis of ship $?x$ is not less than the lower bound $?lower$, and $pre_\perp(a)$ is a set of propositional preconditions, e.g., "(sample-taken ?region)" indicating ROV has taken sample at region "?region". Similarly, we denote $eff(a)$ by $eff(a) = eff_\leftrightarrow(a) \cup eff_\perp(a)$, where $eff_\leftrightarrow(a)$ is a set of numeric effects, e.g., "(increase ?x (* ?v ?t))" indicating that the x-axis of ship $?x$ increases by the product of velocity $?v$ and time $?t$, and $eff_\perp(a)$ is a set of propositional effects, e.g., "(image-taken ?region)" indicating that an image is taken by AUV at region $?region$ after action $a$ is executed. An action model is *discrete* if and only if it has only propositional effects, i.e., $eff_\leftrightarrow(a) = \emptyset$. An action model is *continuous* if and only if it has numeric effects, i.e., $eff_\leftrightarrow(a) \neq \emptyset$. We use $\vec{v}$ to denote the value vector of all numeric parameters occurring in action models.

We denote the cost of action $a$ by $\psi(a)$, which can be defined based on the resource consumption of the action. In this paper, if $a$ is a continuous action we define $\psi(a)$ as the moving distance of an object (e.g., ship); if $a$ is a discrete action, we define $\psi(a)$ as 1.

$s_0 \in \mathcal{S}$ is an initial state. $g \subseteq \mathcal{S}$ is a goal to be achieved. Considering different sizes of obstacles in our planning problem, we denote an obstacle $O$ by $\varphi(L)$ which is an area surrounded by a set of vertexes $L$.

Our problem can be formulated by, given as input a planning problem $\mathcal{M}$, our approach outputs a solution plan $p = \langle a_1, a_2, \ldots, a_n \rangle$ with a minimal cost $C = \sum_i \psi(a_i)$, which achieves $g$ starting from $s_0$. An example of our problem can be found from Figure 2, where Figure 2(a) is a set of action models, where $deploy\text{-}ROV$ is a discrete action and the other one is a continuous one. Figure 2(b) is a goal to be achieved, and Figure 2(c) is an initial state. Note that there are propositions and assignments of variables, e.g., "(equip (ship AUV))" is a proposition indicating AUV is equipped on the ship, and "(= location-$x_{ship}$ 1)" is an assignment indicating the value of variable "location-$x_{ship}$" is 1, "(= $O_1$ $\varphi((4.2, 9.3)(4.6, 10)(6.4, 8.7)(4.2, 9.3)))$" is another assignment indicating the value of obstacle variable $O_1$ is the area defined by $\varphi((4.2, 9.3)(4.6, 10)(6.4, 8.7)(4.2, 9.3))$, which is a triangle (denoted by "$O_1$") as shown in Figure 1. Figure 2(d) is a solution plan to the problem, which is a sequence of discrete or continuous actions.

## Our mxPlanner Approach

### Overview of mxPlanner

The overview framework of mxPlanner is shown in Figure 3, which includes a heuristic module, a transition module, and a loss module. We use $a_i$, $\vec{v}_i$ and $s_i$ to denote the action executed, the value vector of $\vec{v}$ and the state in the step $i$. The target is to find a solution plan $\langle a_0, a_1, \ldots, a_{N-1} \rangle$ with parameters $V = \langle \vec{v}_0, \ldots, \vec{v}_{N-1} \rangle$ which makes the total cost minimal. Given $s_0$ and $g$, the heuristic module predicts an action sequence $p_0$. $s_1$ can be updated by the transition module with a state $s_0$, an action $a_0$ whose numeric parameters are assigned by $\vec{v}_0$. After that, the loss $\mathcal{L}_0$ is calculated by Equation (2) with taking as input $s_1$, $a_0$ and $\vec{v}_0$. In every step $i$, we compute the loss $L_i$ caused by selecting an action $a_i$. We accumulate the loss of every step as $\mathcal{L}_{all}$. Then we minimize $\mathcal{L}_{all}$ by updating parameters $V = \langle \vec{v}_0, \ldots, \vec{v}_{N-1} \rangle$ vis gradient descent. When an obstacle-free plan is executable in $s_0$ with achieving the goal, we find a solution plan with the total cost minimized.

The algorithm of mxPlanner is shown in Algorithm 1,

we first initialize parameters $V$ randomly (line 1). A predicted action sequence $\sigma_i$ is computed by the heuristic module $\mathcal{H}(s_i, g, A)$ (line 5). The next state $s_{i+1}$ is updated by transition module (line 7). The accumulated loss $\mathcal{L}_{all}$ is computed by Equation (6) for a plan $\xi$ with length $N$ (line 10). Parameters $V$ is updated by minimizing loss $\mathcal{L}_{all}$ until $\mathcal{L}_{stop} = 0$ (line 11).

---

**Algorithm 1** `mxPlanner`

---

**input:** $\mathcal{M} = \langle \mathcal{S}, \mathcal{A}, s_0, g \rangle$.
**output:** $\xi$.

1: initialize numeric parameters of actions $V = \langle \vec{v}_0, \ldots, \vec{v}_{N-1} \rangle$ randomly;
2: **while** $\mathcal{L}_{stop} \neq 0$ (Equation (8)) **do**
3:     $\xi = \langle \rangle$;
4:     **while** i = 0, ..., N-1 **do**
5:         predict an action sequence $\sigma_i = \mathcal{H}(s_i, g, A)$;
6:         $\bar{a}_i$ is the first action of $\sigma_i$ and $a_i$ is the origin action of $\bar{a}_i$ before discretization;
7:         update $s_{i+1}$ with $a_i$, $v_i$ and $s_i$ (Equation (1));
8:         $\xi = [\xi | a_i]$;
9:     **end while**
10:    calculate accumulated loss $\mathcal{L}_{all}$ (Equation (6));
11:    update $V$ by minimizing $\mathcal{L}_{all}$ (Equation (7));
12: **end while**
13: return $\xi$;

---

## Heuristic Module

Given a state $s_i$ and goal $g$, the heuristic module aims to estimate an action sequence $\sigma_i$ by "discretizing" continuous actions, where $\bar{a}_i$ is either a discrete or discretized action. An overview of the heuristic module is as shown in Algorithm 2.

---

**Algorithm 2** The heuristic module $\sigma_i = \mathcal{H}(s_i, g, \mathcal{A})$

---

**input:** $s_i$, $g$, $\mathcal{A}$
**output:** $\sigma_i$

1: $\bar{\mathcal{A}} = \emptyset$, $\mathcal{B} = \emptyset$;
2: **for** each numeric variable $x$ **do**
3:     Compute bounds: $\mathcal{B}' = \text{ComputeBounds}(x, \mathcal{A})$;
4:     $\mathcal{B} = \mathcal{B} \cup \mathcal{B}'$;
5: **end for**
6: **for** each action $a \in \mathcal{A}$ **do**
7:     **if** $a$ is a discrete action **then**
8:         $\bar{\mathcal{A}} = \bar{\mathcal{A}} \cup \{a^+\}$;
9:     **else if** $a$ is a continuous action **then**
10:        Discretize $a$: $\bar{a} = \text{DiscretizeAction}(a, \mathcal{B})$
11:        $\bar{\mathcal{A}} = \bar{\mathcal{A}} \cup \{\bar{a}^+\}$;
12:     **end if**
13: **end for**
14: Compute a plan: $\sigma_i = \text{Solve}(\bar{s}_i, g, \bar{\mathcal{A}})$;

---

In previous approaches, heuristics are computed based on their fixed continuous effects or a maximum and minimum of continuous effects, such as Metric-FF (Hoffmann 2003) and ScottyActivity (Fernández-González, Williams, and Karpas 2018). These approaches cannot handle our missions where

action effects are a mix of logical operations and indefinite numeric updating. To predict a relaxed plan, we invoke FF planner (Hoffmann 2001) to delete-free relaxed plans by discretizing the numeric effects into intervals.

1. To ensure our plans will stop as long as the goal is achieved, we add an extra $end$ action $\langle end, pre_g, \emptyset \rangle$ into action set $\mathcal{A}$ where $pre_g = g$ and $\emptyset$ indicates action $end$ has no effect.

2. Given a set $\mathcal{A}$ of action models, for a numeric variable $x$, we define the set of action models as $\mathcal{A}_x$, each of which has at least one precondition involving $x$. Then each action model $a \in \mathcal{A}_x$ includes an upper bound $u_x$ and/or a lower bound $l_x$ for $x$. We collect these bounds and order them accendingly $B_x = b_0 < b_1 < \cdots < b_h$. For each bound $b_k$ we construct a proposition "$(\geq ?x \; b_k)$", implemented by "ComputeIntervals" (line 2 to 5).

3. Because classical planner cannot handle continuous actions, we discretize every continuous action model $a$ by replacing its effect with a numeric effect for building conditions to satisfy preconditions of succeeding actions in a plan, implemented by "DiscretizeAction". Formally, "(increase ?x $\epsilon$)" or "(decrease ?x $\epsilon$)", where $\epsilon$ is any expression, is replaced with "$\{(\geq ?x \; b_0), \ldots, (\geq ?x \; b_h)\}$" and the resulting action is denoted by $\bar{a}$. Intuitively, as we do not know the value of $\epsilon$, we relax the action effect to satisfy all numeric preconditions about $x$.

4. For every discrete or discretized action model $a$, we remove its negative effect and use $a^+$ to denote the resulting action model. We use $\bar{\mathcal{A}}$ denote the set of all these action models (line 6 to 13). The behind idea is to quickly find an action sequence to the goal without considering the negative effects, which is possibly executable.

5. To remove numeric variables, we use propositions "$(\geq ?x \; b_k)$" to replace the value of $?x$. Specifically, for the current state $s_i$, we add the corresponding propositions "$(\geq ?x \; b_k)$" according to the value of $?x$ in $s_i$, and remove all numeric variables. We use $\bar{s}_i$ to denote the resulting state.

6. We invoke FF planner to compute a delete relaxed plan $\sigma_i$ for the planning problem with the current state $\bar{s}_i$, goal $g$ and the expanded action set $\bar{\mathcal{A}}$, by "Solve" (line 14).

In the example of the AUV domain, "ROV-take-sample(B)" and "ROV-take-sample(C)" are two action models in $\mathcal{A}$ whose precondition include numeric variable "location-x". "ROV-take-sample(B)" requires $5 \leq$ location-x $\leq 7.2$ while "ROV-take-sample(C)" requires $9.5 \leq$ location-x $\leq 11$. Then we construct a delete-free copy of "ROV-take-sample(B)$^+$" by requiring only $5 \leq$ location-x. Next, we construct its ordered bound list $B_{\text{location-x}} = 5 < 7.2 < 9.5 < 11$. Consider in Figure 2, the continuous action model "ROV-navigate($?v_x$, $?v_y$,$?t$)" with an effect '(increase location-x (∗ $?v_x$ ?$t$))", now we discretize it into a discrete action model "$\overline{\text{ROV-navigate}}$" with positive effects "$\{(\geq$ location-x 5), $(\geq$ location-x 7.2), $(\geq$ location-x 9.5), $(\geq$ location-x 11)$\}$".

## Transition Module

The transition module aims to obtain the next state $s_{i+1}$ given an action $a$, parameters $v_i$ and a state $s_i$. When the heuristic

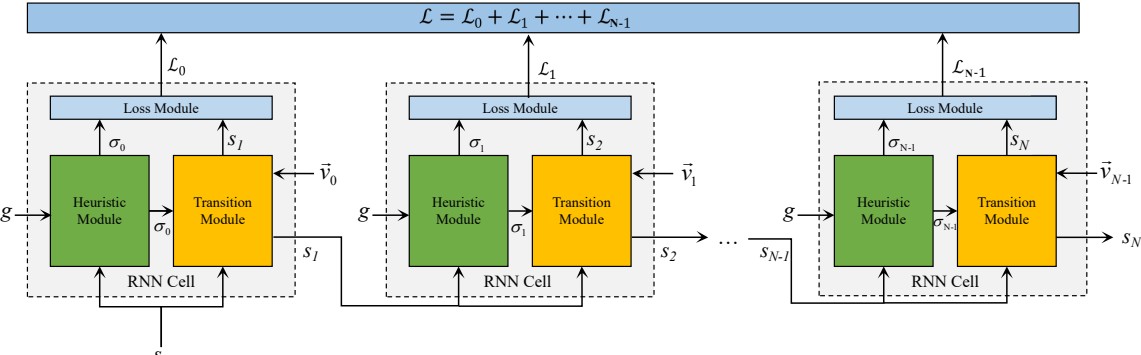

Figure 3: An overview of our model.

module predicts an action sequence $\sigma_i = \langle \bar{a}_0, \ldots, \bar{a}_m \rangle$, we select the original action $a_0$ of the first action $\bar{a}_0$ on the state $s_i$. The state $s_i$ is updated to $s_{i+1}$:

$$s_{i+1} = \gamma(v_i, s_i; \mathit{eff}(a_0)) \qquad (1)$$

In the ocean mission example in Figure 2, a ship navigates by the guidance of x-velocity, y-velocity, and time. An effect of "ROV-navigate($?v_x$-$\mathcal{R}$, $?v_y$-$\mathcal{R}$,$?t$-$\mathcal{R}$)" is "(increase location-x ($* ?v_x ?t$))" indicating that the next x-axis of the robot is the product of $?v_x$ and $?t$ plus currently x-axis. Assuming that the axis of robot location-x = 0, the x-axis of ROV is updated to 2 after action "ROV-navigate(2, -2, 1)". Also, "(equip (ship ROV))" is updated to "(not (equip (ship ROV)))" after action "deploy-ROV" with an effect "(not (equip (ship ROV)))".

## Planning through Gradient Descent

Next, we aim to inversely optimize the input of our RNN framework, i.e., parameters $V$, assuming the model of RNN is provided (i.e., the heuristic and transition modules), which is different from previous RNN approaches to time series predictions (Rumelhart, Hinton, and Williams 1986; Pascanu, Mikolov, and Bengio 2013) via learning model parameters given input data. Our work is similar to previous work done by (Wu, Say, and Sanner 2017), which aims to calculate continuous action sequences via optimizing the input of a given RNN. We extend the work to computing plans by considering logical relations.

In every step, we design a novel loss function by three losses, as shown in Equation (2):

$$\mathcal{L}_i = w_1 \mathcal{L}_{b_i} + w_2 \mathcal{L}_{o_i} + w_3 \psi(a_i) \qquad (2)$$

where $w_1, w_2$ and $w_3$ are weights for the three losses as hyperparameters.

First, $\mathcal{L}_{b_i}$ is a loss to satisfy numeric bounds of action sequence to be executed which is predicted by heuristic module. This loss is defined by Equation (3),

$$\mathcal{L}_{b_i} = ||\mathrm{ReLU}(\vec{s}_{i+1} - \vec{u})||_2 + ||\mathrm{ReLU}(\vec{l} - \vec{s}_{i+1})||_2 \qquad (3)$$

where $\mathrm{ReLU}(x) = \max(0, x)$. For numeric variable $x$, let $\bar{a}_k$ be the first action in the action sequence $\sigma_i$ predicted by $\mathcal{H}(s_i, g, \mathcal{A})$, whose precondition includes $x$. It requires $x$ to

fall within the interval $[l_x, u_x]$. We use $\vec{l}$ and $\vec{u}$ to denote all bounds for all numeric variables. Once a numeric variable $x$ in the state $s_{i+1}$ exceeds its upper bound $u_x$, i.e., $x - u_x > 0$, a loss is generated. The case for lower bounds is similar.

Second, $\mathcal{L}_{o_i}$ is a loss to avoid obstacles, defined by:

$$\mathcal{L}_{o_i} = \sum_{\alpha=1}^{M} m_\alpha ||y'_\alpha - p_{i+1}||_2 \qquad (4)$$

where $m_\alpha$ is defined by:

$$m_\alpha = \begin{cases} 1, & \text{if the line between } p_i \text{ and } p_{i+1} \\ & \quad \text{intersects with } O_\alpha \\ 0, & \text{otherwise} \end{cases} \qquad (5)$$

$p_i$ is the position of the robot in the state $s_i$, $x'_\alpha$ is a selected aim position which guides the robot to avoid the obstacle $O_\alpha$, $\alpha = 1, \ldots, M$. First, We use $Y_\alpha$ to denote the set of vertexes $y$ of obstacle $O_\alpha$ such that the line between $y$ and $p_i$ avoids $O_\alpha$. Then we define $y_\alpha$ as the closest vertex to $p_{i+1}$ in $Y_\alpha$, i.e., $y_\alpha = \mathrm{argmin}_{y \in Y_\alpha} ||p_{i+1} - x_\alpha||_2$. Next, to lead avoiding the obstacle $O_\alpha$, we define $x'_\alpha$ as a position which satisfies $||x'_\alpha - x_\alpha||_2 = \varepsilon$ where $\varepsilon$ is a small positive real number and the line between $x'_\alpha$ and $p_i$ avoids $O_\alpha$. An example is shown in Figure 4. Intuitively, when the robot tries to go through the obstacle, $L_{oi}$ aims to guide its destination $p_{i+1}$ getting close to $y'_\alpha$ for getting rid of $O_\alpha$.

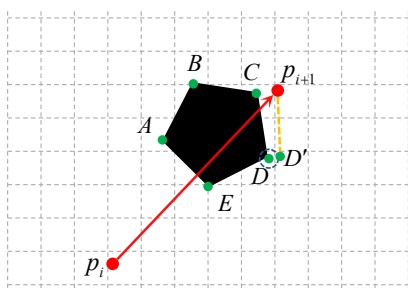

Figure 4: The black pentagon ABCDE is an obstacle $O_\alpha$, $p_i$ is the current position and $p_{i+1}$ is the expected next position. $Y_\alpha = \{A, D, E\}$ and $y_\alpha = D$. $D'$ is a position outside $O_\alpha$ that is $\varepsilon$ from $D$ and is a candidate $y'_\alpha$.

Third, $\psi(a_i)$ is the cost of $a_i$. For example, we can consider action costs as the navigating distance: the effects of "ROV-navigate($?v_x$, $?v_y$,$?t$)" are "(increase location-x (* $?v_x$ $?t$))", "(increase location-y (* $?v_y$ $?t$))" and "(increase total-time $?t$)", so its cost is $\sqrt{(tv_x)^2 + (tv_y)^2}$.

We define the accumulated loss $\mathcal{L}_{all}$ as the sum of instantaneous losses until the goal is achieved, i.e.,

$$\mathcal{L}_{all} = \sum_{i=0}^{\mu-1} \mathcal{L}_i, \quad \text{s.t. } a_\mu = end \quad (6)$$

Then we compute the partial derivatives of the accumulated loss as gradients. Formally, $v_i$ is a numeric parameter of $a_i$ at the $i$-th step.

$$
\begin{aligned}
\frac{\partial \mathcal{L}_{all}}{\partial v_i} &= \sum_{\lambda=i}^{\mu-1} \frac{\partial \mathcal{L}_\lambda}{\partial v_i} \\
&= \sum_{\lambda=i}^{\mu-1} \left( \frac{\partial \mathcal{L}_{b_\lambda}}{\partial v_i} + \frac{\partial \mathcal{L}_{o_\lambda}}{\partial v_i} + \frac{\partial \psi(a_\lambda)}{\partial v_i} \right) \quad (7) \\
&= \sum_{\lambda=i}^{\mu-1} \left( \frac{\partial \mathcal{L}_{b_\lambda}}{\partial \vec{s}_{\lambda+1}} \frac{\partial \vec{s}_{\lambda+1}}{\partial v_i} + \frac{\partial \mathcal{L}_{o_\lambda}}{\partial \vec{s}_{\lambda+1}} \frac{\partial \vec{s}_{\lambda+1}}{\partial v_i} \right) + \frac{\partial \psi(a_i)}{\partial v_i}
\end{aligned}
$$

Intuitively, the gradient of $v_i$ is determined by the states from $s_{i+1}$ to $s_\mu$ and the cost of action $a_i$. The gradient of the numeric parameters irrelated to $a_i$ is zero. By gradient descent, the total cost as a loss is minimized.

To guarantee that the action sequence $\xi$ is executable, achieve the goal, avoid all obstacles, we define $\mathcal{L}_{stop}$ in Equation (8). When $\mathcal{L}_{stop} = 0$, a solution plan is found.

$$\mathcal{L}_{stop} = \begin{cases} \sum_{i=0}^{\mu} \mathcal{L}_{o_i}, & \text{if } g \subseteq s_{\mu+1} \text{ and} \\ & \quad \xi \text{ is executable} \quad (8) \\ \infty, & \text{otherwise} \end{cases}$$

## Properties

Our mxPlanner approach has the following *soundness* and *completeness* properties.

**Soundness:** The action sequence computed by mxPlanner is a solution plan for the planning problem.

*Sketch of proof:* When mxPlanner outputs a plan, according to line 2 in Algorithm 1, the loss $\mathcal{L}_{stop} = 0$. In other words, all obstacles are avoided when reaching the goal, and the plan is executable, i.e., preconditions of actions are satisfied at the states where they are executed. Thus, the output action sequence is a solution plan for the problem.

**Completeness:** Our mxPlanner approach is complete if the following conditions are satisfied: (1) the planning problem is solvable; (2) the length $N$ in Algorithm 1 is finite but large enough for solving the planning problem; (3) the off-the-shelf planner that we utilize in mxPlanner is complete.

*Sketch of proof:* If the off-the-shelf planner is complete, i.e., Step 14 of Algorithm 2 will output a plan $\sigma_i$ and the

original action $a_i$ of the first action $\bar{a}_0$ of $\sigma_i$ should be executable in state $s_i$, the resulting plan $\xi$ computed by line 4-9 in Algorithm 1 is possibly executable starting from $s_0$ and eventually reaches $g$ since $N$ is large enough for solving the problem. With line 10-11 in Algorithm 1, $\mathcal{L}_{all}$, as well as $\mathcal{L}_{stop}$, will be reduced with gradient descent at each repetition (line 2-12) until the stop condition is satisfied and Algorithm 1 outputs the solution plan $\xi$, which indicates our mxPlanner is complete.

## Experiments

To evaluate the performance of our approach, we demonstrate our results in the AUV domain and the Taxi domain with comparison with Metric-FF (Hoffmann 2003) and ScottyActivity (Fernández-González, Williams, and Karpas 2018).

### The AUV Domain

In this domain, an AUV (automated underwater vehicle) needs to reach goal regions and take samples under the condition of avoiding obstacles. We modified the AUV domain used in ScottyActivity (Fernández-González, Williams, and Karpas 2018) with two main changes. Firstly, effects of the *glide* action are computed by three variables, x-velocity $v_x$, y-velocity $v_y$ and execution time of action $t$, such as "increase x-location (* $v_x$ $t$)". Owing to neither Metric-FF nor ScottyActivity can handle variable flexible continuous effects with numeric parameters, we randomly set several groups of parameters. For space limitation, we only show some of them. Secondly, obstacles are also added to the domain randomly.

The optimization objective for this domain is navigating distance. In this domain, only moving actions such as "glide($?t$, $?v_x$, $?v_y$)" generates costs. Considering its effects: "increase x-location (* $?v_x$ $?t$)" and "increase y-location (* $?v_y$ $?t$)". Its cost is $\sqrt{(tv_x)^2 + (tv_y)^2}$. We set the weights $w_1 = w_2 = w_3 = 1$ and the distance threshold $\varepsilon = 0.1$.

To show defects of fixed continuous effects, Table 1 shows navigating distance of obstacle-free plans between mxPlanner and Metric-FF with two groups of parameters, and performance of mxPlanner compared with ScottyActivity in handling no obstacle problems. Comparing with column "Metric-FF ($t = 1$, $v_x = 20$, $v_y = 20$)" and column "Metric-FF ($t = 1.5$, $v_x = 20$, $v_y = 20$)", most of problems could not be solved when $t = 1.5$, $v_x = 20$, $v_y = 20$ which is denoted by "\" which means the problem could not be solved in 1 hour. The reason of no solution is Metric-FF under this parameter setting could not find a plan arriving the bounds of some regions need to be taken samples. However, Metric-FF with $t = 1.5$, $v_x = 20$, $v_y = 20$ performs better in problem 11 and 12. It shows that it is not easy to choose a fit continuous effect manually. Unfit fixed continuous effects set in advance result in more consumption or no solution. mxPlanner performs better because mxPlanner optimize parameters by gradient descent instead of assignment manually. In brief, fixed continuous effect is inefficient and fallible compared with variable parameters.

The comparison on navigating distance between ScottyActivity and mxPlanner in 20 problems without obstacles is shown in the last two columns of Table 1. mxPlanner

Table 1: Comparison among `mxPlanner`, Metric-ff and ScottyActivity with different fixed parameters $t, v_x, v_y$ in 20 problems. Each problem has at most five target sample regions need to be arrived.

| | `mxPlanner` ($t \geq 0, v_x, v_y \in \mathcal{R}$) | | | | Metric-FF ($t=1, v_x=20, v_y=20$) | | | | Metric-FF ($t=1.5, v_x=20, v_y=20$) | | | | `mxPlanner` ($0 \leq t \leq 1$ $|v_x|,|v_y| \leq 10$) | ScottyActivity ($0 \leq t \leq 1$ $|v_x|,|v_y| \leq 10$) |
|----|--------|--------|--------|--------|--------|--------|--------|--------|--------|--------|--------|--------|--------|--------|
| | 0 | 1 | 2 | 4 | 0 | 1 | 2 | 4 | 0 | 1 | 2 | 4 | | |
| 1 | 106.31 | 106.79 | 108.19 | 108.92 | 113.14 | 124.85 | 124.85 | 124.85 | \ | \ | \ | \ | 106.68 | 106.46 |
| 2 | 123.27 | 123.79 | 127.29 | 127.29 | 144.85 | 156.57 | 156.57 | 156.57 | \ | \ | \ | \ | **123.14** | 125.76 |
| 3 | 109.58 | 111.71 | 118.93 | 118.93 | 136.57 | 148.28 | 148.28 | 148.28 | \ | \ | \ | \ | **109.30** | 114.99 |
| 4 | 186.07 | 187.36 | 187.53 | 187.53 | 236.57 | 236.57 | 236.57 | 442.84 | \ | \ | \ | \ | **184.60** | 249.94 |
| 5 | 195.82 | 200.96 | 204.99 | 218.36 | 241.42 | 281.42 | 281.42 | 377.99 | \ | \ | \ | \ | **200.50** | 284.94 |
| 6 | 126.52 | 126.56 | 126.53 | 129.19 | 136.57 | 136.57 | 136.57 | 136.57 | \ | \ | \ | \ | **128.01** | 158.47 |
| 7 | 94.35 | 95.44 | 95.77 | 95.97 | 104.85 | 104.85 | 116.57 | 116.57 | 114.85 | 132.43 | 132.43 | 132.43 | **94.35** | 96.34 |
| 8 | 171.52 | 172.86 | 177.08 | 174.27 | 193.14 | 193.14 | 193.14 | 216.57 | \ | \ | \ | \ | **169.94** | 229.94 |
| 9 | 166.77 | 167.95 | 170.71 | 170.87 | 181.42 | 204.85 | 204.85 | 216.57 | 199.71 | 217.28 | 217.28 | 234.85 | 167.30 | **166.95** |
| 10 | 141.74 | 142.83 | 143.54 | 155.74 | \ | \ | \ | \ | \ | \ | \ | \ | **141.73** | 175.00 |
| 11 | 149.30 | 149.30 | 149.30 | 152.98 | 322.84 | 277.99 | 329.71 | 477.99 | 169.71 | 187.28 | 187.28 | \ | **149.93** | 152.04 |
| 12 | 213.52 | 216.02 | 217.12 | 220.15 | 334.56 | 399.41 | 442.84 | 477.99 | 247.28 | 247.28 | 247.28 | \ | **215.16** | 325.65 |
| 13 | 223.70 | 223.87 | 241.95 | 265.53 | 249.71 | 249.71 | 249.71 | 281.42 | \ | \ | \ | \ | 226.66 | **221.86** |
| 14 | 265.61 | 265.90 | 280.18 | 286.48 | 306.27 | 306.27 | 306.27 | 434.56 | \ | \ | \ | \ | **318.15** | 372.25 |
| 15 | 195.70 | 254.18 | 254.24 | 288.22 | 431.13 | 602.84 | 640.83 | 642.84 | \ | \ | \ | \ | **195.70** | 314.00 |
| 16 | 77.04 | 77.04 | 77.04 | 84.6 | 96.57 | 96.57 | 96.57 | 108.28 | 102.43 | 102.43 | 102.43 | \ | **77.12** | 79.30 |
| 17 | 96.43 | 96.44 | 96.44 | 99.46 | 136.57 | 136.57 | 136.57 | 148.28 | \ | \ | \ | \ | 97.10 | **96.60** |
| 18 | 138.57 | 138.89 | 138.89 | 147.24 | 184.85 | 184.85 | 184.85 | 228.28 | \ | \ | \ | \ | **145.52** | 201.38 |
| 19 | 201.49 | 202.00 | 227.58 | 267.54 | 241.42 | 241.42 | 273.14 | 308.28 | \ | \ | \ | \ | **202.52** | 322.83 |
| 20 | 216.01 | 224.74 | 235.73 | 243.39 | 487 | 515.98 | 557.98 | 671.13 | \ | \ | \ | \ | **236.11** | 387.26 |

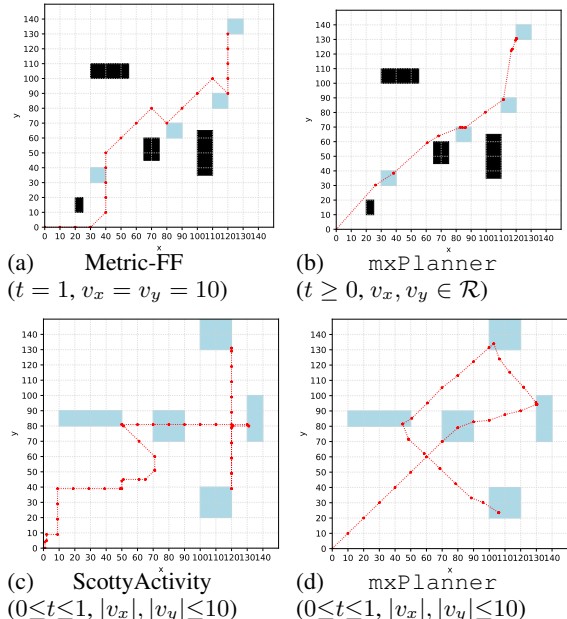

(a) Metric-FF ($t=1, v_x=v_y=10$)

(b) `mxPlanner` ($t \geq 0, v_x, v_y \in \mathcal{R}$)

(c) ScottyActivity ($0 \leq t \leq 1, |v_x|,|v_y| \leq 10$)

(d) `mxPlanner` ($0 \leq t \leq 1, |v_x|,|v_y| \leq 10$)

Figure 5: Plans of Metric-FF, ScottyActivity and `mxPlanner`. Red dots are locations of robots which make a trajectory. Blue areas are target sample regions and black areas are obstacles.

performs better than ScottyActivity except for Problem 1,9,13,16. An example is shown in Figure 5(c)(d), the former is based on ScottyActivity and the latter is computed by `mxPlanner`. Each movement of Figure 5(c) is also uneven owing to that ScottyActivity make use of convex optimization to optimize an ordered action sequence calculated in advance. To compute an action sequence, ScottyActivity's continuous updatings also need fixed parameters, or maximum and minimum for heuristic searching. The effectivity of `mxPlanner` is due to that `mxPlanner` is computed at each step based on current state instead of computing a whole plan in advance and `mxPlanner` has no need for manually assignments for heuristic searching. However, solution of `mxPlanner` may be not optimal. Compared with Scotty-Activity, `mxPlanner` is not limited to convex problems. To handle problems with obstacles, ScottyPath is based on ScottyActivity with generating safe regions in advance by giving up parts of areas. `mxPlanner` makes no use of deleting searching areas, hence `mxPlanner` avoids situations which may discard solutions in the beginning.

## The Taxi domain

In this section, we evaluate our approaches on a domain – Taxi domain. In this domain, an agent needs to drive a car to pick up passengers who may move and then to carry them to their destination. Movement of the car is still computed by three parameters, x-velocity $v_x$, y-velocity $v_y$ and action execution time $t$. A passenger can be picked up when the car nears him. Movement of a passenger is captured by a predefined function w.r.t. time $t$ and velocity of passengers $v_p$. An example is shown in Figure 6, the car needs to pick up two passengers and carry them to region A. Initially, passenger locations are in the gray human forms ($P_1$ when $t=0$ and $P_2$ when $t=0$) and they move along with the trajectoryy of

Table 2: Navigating distance of 10 plans in simple taxi domain without passenger movements

| | mxPlanner $(0 \leq t \leq 1,$ $|v_x| \leq 10,$ $|v_y| \leq 10,$ | mxPlanner $(0 \leq t \leq 1,$ $|v_x| \leq 20,$ $|v_y| \leq 20,$ | mxPlanner $(0 \leq t \leq 1,$ $|v_x| \leq 30,$ $|v_y| \leq 30,$ | ScottyActivity $(0 \leq t \leq 1,$ $|v_x| \leq 10,$ $|v_y| \leq 10,$ | ScottyActivity $(0 \leq t \leq 1,$ $|v_x| \leq 20,$ $|v_y| \leq 20,$ | ScottyActivity $(0 \leq t \leq 1,$ $|v_x| \leq 30,$ $|v_y| \leq 30,$ | Metric-FF $(t = 1,$ $v_x = 5,$ $v_y = 5,$ | Metric-FF $(t = 1,$ $v_x = 10,$ $v_y = 10,$ | Metric-FF $(t = 1,$ $v_x = 20,$ $v_y = 20,$ | Metric-FF $(t = 1,$ $v_x = 30,$ $v_y = 30,$ |
|---|---|---|---|---|---|---|---|---|---|---|
| 1 | 143.24 | 141.59 | 141.58 | 198.71 | 199.18 | 199.82 | 203.64 | 141.42 | 141.42 | \ |
| 2 | 109.37 | 114.17 | 114.17 | 137.07 | 139.65 | 134.11 | 139.50 | 120.71 | 156.57 | \ |
| 3 | 113.14 | 113.14 | 113.14 | 157.07 | 159.77 | 159.82 | 194.85 | 176.57 | 113.14 | 127.28 |
| 4 | 128.02 | 127.73 | 127.73 | 168.42 | 169.18 | 168.21 | 276.07 | 214.85 | 237.99 | 264.85 |
| 5 | 128.06 | 127.66 | 127.65 | 178.48 | 179.06 | 177.86 | 127.28 | 257.28 | 141.42 | \ |
| 6 | 128.77 | 127.83 | 127.83 | 178.48 | 179.06 | 177.86 | 127.28 | 200.71 | 141.42 | \ |
| 7 | 166.82 | 166.83 | 167.24 | 220.90 | 223.48 | 221.96 | 264.35 | 243.14 | 584.26 | \ |
| 8 | 127.58 | 127.65 | 127.70 | 159.24 | 179.06 | 177.82 | 127.28 | 285.56 | 346.27 | \ |
| 9 | 162.20 | 162.49 | 162.37 | 218.71 | 217.30 | 217.72 | 641.13 | 633.55 | 707.70 | \ |
| 10 | 200.50 | 206.67 | 204.00 | 256.76 | 245. 72 | 256.06 | 301.42 | 305.56 | 353.14 | \ |

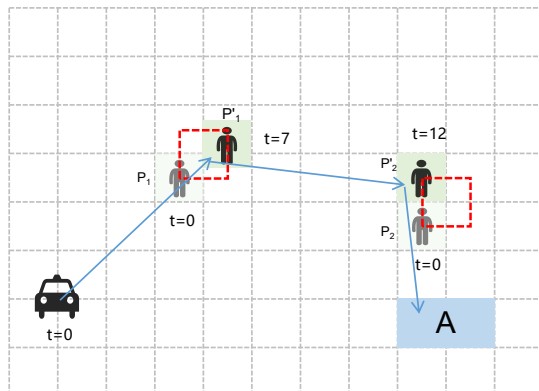

Figure 6: An example of taxi domain, the car needs to pick up two passengers in green areas and take them to region A. The initial location of passengers are the gray human forms and passengers move along with the trajectory of red squares with the total time $t$ growing, then they reach new locations (the black human forms) and board the car.

red squares. The car picks up the first passenger when $t = 7$ at $P'_1$ and picks up the second passenger when $t = 12$ at $P'_2$.

However, Metric-FF cannot handle this domain and we provide a simple version without passengers' movement, i.e., assume every passenger is located in the initial state. As for ScottyActivity, passenger movement trajectories are not convex, which leads it difficult to apply directly ScottyActivity. To do so, we first assume every passenger keep standing and compute an action sequence. Then parameter $t$ for every action is fixed. It determines the location of passengers, which allows us to construct a convex problem to optimize the navigation distance via varying $v_x$ and $v_y$.

In this domain, optimization objective is navigating distance which is defined as $\sqrt{(tv_x)^2 + (tv_y)^2}$ in AUV domain. It is also generated by action "glide($t, v_x, v_y$)" whose effects are "increase x-location (* ?$v_x$ ?$t$)" and "increase y-location (* ?$v_y$ ?$t$)". Similarly, We set the weights $w_1 = w_2 = w_3 = 1$ and velocity of passengers $v_p = 1.25$.

Table 3: Navigating distance of 10 plans in taxi domain

| | mxPlanner $(0 \leq t \leq 1,$ $|v_x| \leq 10,$ $|v_y| \leq 10)$ | ScottyActivity $(0 \leq t \leq 1,$ $|v_x| \leq 10,$ $|v_y| \leq 10)$ | mxPlanner $(0 \leq t \leq 1,$ $|v_x| \leq 20,$ $|v_y| \leq 20)$ | ScottyActivity $(0 \leq t \leq 1,$ $|v_x| \leq 20,$ $|v_y| \leq 20)$ |
|---|---|---|---|---|
| 1 | 141.63 | 198.71 | 141.51 | \ |
| 2 | 117.83 | \ | 126.85 | \ |
| 3 | 113.20 | \ | 113.14 | \ |
| 4 | 140.79 | 168.42 | 152.11 | \ |
| 5 | 127.68 | 178.48 | 128.44 | \ |
| 6 | 128.29 | 178.48 | 130.03 | 179.06 |
| 7 | 173.82 | 220.90 | 178.594 | \ |
| 8 | 128.33 | 178.48 | 129.25 | 179.06 |
| 9 | 163.84 | \ | 171.33 | \ |
| 10 | 213.30 | \ | 220.01 | \ |

Table 2 shows the navigating distances of 10 problems where passenger locations are assumed to be fixed for three approaches. Similarly, we set four groups of parameters for Metric-FF and three groups of parameters for ScottyActivity. Compared with column "Metric-FF" with four groups of parameters, where "\" means that Metric-FF could not find a solution in 1 hour, Metric-FF with $t = 1, v_x = v_y = 5$ compute plans with the least cost in problem 5,6,8 and 10 in serval cases. And Metric-FF with $t = 1, v_x = v_y = 10$ performs better than the others in problem 1,2,4,7 and 9. But Metric-FF with $t = 1, v_x = v_y = 30$ could only solve problem 3 and problem 4. As for ScottyActivity, compared with column "ScottyActivity" with three groups of parameters, ScottyActivity with $0 \leq t \leq 1, |v_x|, |v_y| \leq 30$ generates plans with less cost in problem 2,4,5 and 6, and ScottyActivity with $0 \leq t \leq 1, |v_x|, |v_y| \leq 10$ performs better in problem 3,7 and 8. The results also shows that it is hard and inefficient to set suitable fixed parameters in advance. And mxPlanner computes a plan by gradient descent instead of parameters assignment manually. All the results confirm our viewpoints, i.e., fixed continuous effect and manual assignment are inefficient and fallible compared with variable parameters.

We also test their performance for the case of moving passengers. Table 3 shows the navigating distances of 10 problems with passenger movements compared with `mxPlanner` and ScottyActivity, we only show two groups of parameters for space limitations. By constraining $|v_x|, |v_y| \leq 10$ and $|v_x|, |v_y| \leq 20$, `mxPlanner` obtains the top performance in the ten problems while ScottyActivity fails to solve four problems within the cut off time. The failure results from the fact that the passengers are not in the initial location and it is impossible to get close to the new passenger location. It also shows that `mxPlanner` is effective to solve non-convex planning problems.

## Conclusion

In this paper, we approach `mxPlanner` which is based on RNNs combined with heuristic searching to handle a mix of discrete and continuous actions with purpose of goal arriving, obstacle avoidance and objectives minimization. Compared with planner Metric-FF and ScottyActivity, `mxPlanner` can handle non-linear continuous effects and obstacles avoidance without fixed continuous effects assigned manually. All the results show that our approach can generate high-quality plans efficiently. In future work, we plan to find more efficient approaches to improve heuristic searching and obstacle avoidance to reduce time-consumption and cost to fulfill real-world applications.

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
