# OpenReview forum: "RNNs Based Planning with Discrete and Continuous Actions"
_icaps-conference.org/ICAPS/2020/Workshop/HSDIP — Submitted to HSDIP 2020_

### Official Review · AnonReviewer1 · 2020-03-31
**Early assessment**

**Rating:** 5
**Confidence:** 2

**Review:**

The topic of this paper is planning with discrete and continuous actions in order to create plans for realistic planning scenarios. In the paper, an algorithm is presented which uses a combination of heuristic search and gradient descent. A theoretical analysis is provided which proves properties of the proposed approach and an empirical analysis is conducted.

The topic of the paper fits to the workshop as the HSDIP workshop explicitly welcomes multidisciplinary work such optimization and the application of machine learning in heuristic search.

Overall, the defined planning problem is interesting, but also challenging. My main feedback is that parts of the paper are difficult to parse and not always easy to understand. However, some things can easily be fixed. Abbreviations are often used in the paper before they are introduced, causing confusion, e.g. RNN, ROV, AUV. Frequently convex and non-convex problems are mentioned, but it is not really defined what a convex and non-convex problem is. Since this plays a central role, it would be advantageous to clearly define what a convex or non-convex problem is. The definition of the initial state shown in Figure 2c does not correspond to the explanations in the text. In Figure 2c the area of the obstacle $O_1$ is defined as $\varphi(( 4.2 , 9.3 ), ( 4.6 , 10 ), ( 6.4, 8.7 ))$, while in the text the area of the obstacle $O_1$ is defined as $\varphi(4.2, 9.3)(4.6, 10)(6.4, 8.7)(4.2, 9.3))$. Given the first and the last point in the second definition are the same, I assume that both define the same triangle?

It is noted that Metric-FF requires human computation in advance. It would help to clarify why Metric-FF requires such a computation in advance, while mxPlanner does not need it, although mxPlanner uses Metric-FF.

The properties section shows that mxPlanner is complete and sound. Since numerical planning is semi-decidable, do these results imply that the presented planning formalism is also semi-decidable?

Finally, in the Heuristic Module section, an interval relaxation is mentioned. I wonder if this is related to work on interval relaxed numeric planning.

Minor Comments:
- RNN cell is used in Figure 3, but it is not explained what a "cell" is in this context.
- In the Problem Definition section $p$ is used for a plan, while $\xi$ is used in Algorithm 1. Is there a reason to use a different notation?
- Algorithm 2 does not return $\sigma_i$
- Citations: e.g. Gracia et al. (Gracia, Sala, and Garelli 2012) -> Gracia et al. (2012)
- ...“vis” gradient descent. -> ...with gradient descent. ?
- First, *We* use ...
- advance.To -> advance. To
- ...with the trajectoryy -> with the trajectory
- ...on a domain – Taxi domain. -> on the Taxi domain.

---

> ### Comment · AnonReviewer1 · 2020-08-17
> **Final remark**
>
> After revisiting the paper, I still think that the points made in my initial assessment should be taken into account and incorporated in order to improve the paper. As I think this is essential for understanding the paper, I am updating the score accordingly. Since there is no new version of the paper or response, I have nothing more to add at this point.

---

### Official Review · AnonReviewer2 · 2020-04-07
**Early assessment**

**Rating:** 7
**Confidence:** 3

**Review:**

The paper presents an RNN approach to solving a class of planning problems with a mix of discrete and continuous actions. The presented approach can handle an extended class of problems, compared with previous approaches, and experiments show that high-quality plans can be obtained efficiently.

The contributions of the paper are relevant to the topic of the workshop. Overall, the technical quality of the paper is good. The paper includes examples and figures, and complements the algorithms with an empirical study. Some parts of the paper need to be rephrased (e.g., several sentences in the experimental section), but in general is well redacted.

The empirical study shows a comparison with existing algorithms on two domains. I missed an additional study of domains or problem sizes that pushes mxPlanner to the limits. Perhaps in certain classes of problems using Metric-FF is preferable, and knowing this in advance will be good to practitioners. Such a study could be done in a future revision of the paper if the authors plan to submit it for presentation as a full conference paper.

I believe that the paper will be more accessible to people with a limited deep learning background if the basics of RNNs are explained. Certainly, planning models are explained in detail, but RNNs are not. The caption of Figure 3 could also be used to summarize the approach, and explain to the reader where the RNN loop is. On a related note, I suggest the authors use the captions in each of the tables to tell the reader what the numbers mean, and make sure this is also said in the text.
In terms of organization, I found it weird that equation numbers were mentioned on page 3 (Section “Our mxPlanner Approach”), but these equations do not appear until much later. Also, Figure 3 is first mentioned in page 3 as well, but appears on page 5.

---

### Official Review · AnonReviewer3 · 2020-04-08
**Early Assessment**

**Rating:** 3
**Confidence:** 4

**Review:**

Brief summary of the paper:
This paper presents a novel approach based for planning problems that include discrete and continous actions. The main idea is to represent the assignment of action parameters as a loss function optimization problem and applying a recurrent neural network to solve this optimization problem.

Major remarks:

- First, I would like to ask the authors why they decided to submit their work to the HSDIP workshop instead of PlanRob. It is true that HSDIP welcomes multidisciplinary work, but the topic of the workshop has a focus on heuristic and search for domain-independent planning, and I would argue that the main topic of this paper is numeric planning with mixed discrete and continous action spaces and is also somewhat tailored to a particular class of problems. The PlanRob workshop explicitly welcomes work on mission, path, and motion planning for robots, so I think that this workshop is a much better fit for the paper than the HSDIP workshop.

- I found the mxPlanner section hard to read because there are so many forward references to equations that are important to understand the algorithm. Additionally, the section is also hard to follow for someone with a weak background on RNNs and reinforcement learning in general. There is no intuition given on why the heuristic and transition model have to be implemented as modules and how such modules interact in general with a recurrent neural network. I think it would be easier of the reader if the subsection on 'Planning through Gradient Descent' is presented first and only focusses on how the assignment of vectors of continous actions can be translated to neural networks by understanding the underlying problem as an optimization problem which minimizes a loss function.

- The first paragraph of the mxPlanner section states that the found plan is one with minimal cost. I don't see why this is true and it is also not part of the Properties section.

Minor comments:

- I suggest you ask someone who is a native english speaker to proof read the paper, as there a couple of grammar errors throughout the paper. Just to list a few things on the first page:
  - '[...], robotics community has made tremendous success with trajectory optimization' -> _the_ robotics community / had tremendous success
  - 'it can not handle missions with short and long-time activities coexist' -> _where_ short and long-time activities co-exist
  - 'can handle a mix of discrete and continous action in convex domain' -> actions

- I think it is easier to follow the paper if the ocean missing scenario is given before the cursive part of the introduction instead of refering to it as 'described below'

- Figure 1 is really hard to read on print-out. Increase the DPI of the figure and use darker shades of green and orange

- In algorithm 1, what is the intuition behind initializing the parameters of V randomly? Could there be a better initialization than random?

---

### Comment · AnonReviewer3 · 2020-08-18

Brief summary of the paper:

This paper presents a learning approach for a restricted class of planning problems that include discrete and continous actions. The main idea is to represent the assignment of action parameters as a loss function optimization problem and use gradient-descent to construct a cost-minimal sequence of actions.

Brief summary of the review:

The paper is hard to follow because the underlying problem definition is not well-defined and leaves many details unclear. While the general idea of the proposed approach becomes apparent, there are many details missing. Additionally, the paper seems to target a specific class of numeric planning problems: navigation problems with obstacles and a restricted set of numerical functions (increase and decrease). Therefore, the question of my initial assessment still stands: why is this not a submission to a workshop that is focussed on planning and robotics? The HSDIP workshop explicitly encourages work on *domain-independent* planning, and I see this approach as a purely domain-dependent algorithm. I therefore update my original assessment and reduce my score from 5 (marginally below) to 3 (clear reject).

Major points:

 - The problem definition of a planning problem seems to mix a lifted planning problem with a grounded planning problem. Additionally, the section lacks many definitions which are necessary to give the problem any meaning:

    - the definition of a planning problem says that a state is composed of a set of propositions, "including numerical equations", but then the value vector is defined over numerical variables. In numeric planning it is more reasonable to differentiate between propositional state variables and numerical state variables, and a numerical equation is an n-ary function f over variables $v_1, \dots, v_n$ and a relational operator.

    - The definition of action models seems to be the lifted action description (also sometimes called 'action schema'). This should be made explicitly clear, because this is not usually part of a 'planning problem', but the instantiated set of actions is (which becomes the set of actions in a planning problem). There is also no definition of what a 'parameter' is, but nevertheless it is required for $\stackrel{\rightarrow}{v}$ to be meaningful.

    - The definition of what a precondition and an effect is is missing. The paper just gives an example of what a precondition looks like, but not a formal definition. Often numerical conditions and effects consist of an n-ary function f over n numerical state variables $v_1, \dots, v_n$ and a relational operator. Even if the exact definition is not relevant for the paper it makes it easier for the reader if the definition of the problem is sound.

    - The paper denotes the 'cost of an action', but an 'action' is not defined, only an action model. That an action is a result of instantiating the action model has to be made explicit.

    - The paper defines the cost of an action as 'as the moving distance of an object'. This is not a formal definition. Why is this different to the usual definition of action cost functions for numerical planning tasks?

    - Suddenly 'obstacles' are part of the planning problem, which are defined as an area surrounded by vertices which are also never formally introduced beforehand.

    - "Our problem can be formulated by, given an input problem M, our approach outputs a solution plan": this is not a problem definition. But I assume that the underlying problem is to compute a cost-minimal plan that achieves the goal starting from the initial state for a given instantiation of a lifted planning problem. Note that without any restrictions on the problem class this is in general undecidable ["Decidablity and Undecidability Results for Planning with Numerical State Variables", Helmert, AIPS 2002]

  - The ongoing intersection between definition and examples makes this section  hard to read. I recommend to just give the formal definition of the problem and then a complete example.

  - Given the lack of precision in the problem definition, the mxPlanner approach description also makes it hard to follow for the reader, as now the approach considers an action together with the vector of parameters. All in all, the reader has to somewhat guess what exactly the underlying problem is.

 - I don't see where N is defined, but N clearly is the number of actions in the plan (i.e. plan length). However, Algorithm 1 starts with initializing the parameters of N actions randomly. This assumes that the plan length is known beforehand.

 - "These approaches cannot handle our missions where action effects are a mix of logical operations and indefinite numeric updating" - what is indefinite numeric updating? Once again, the original problem formulation is just not sufficiently precise to follow the paper

 - comments on the description of the heuristic module:
    - why do we require an extra end action? What does "plans will stop" mean?

    - Formally bounds on variables were never introduced. How are they computed? Where do they come from? Note also that 2. says 'implemented by ComputeInverals', but the function in Algorithm 2 says ComputeBounds.

    - In 3. it says (increase ?x epsilon) and (decrease ?x epsilon), what about other numerical effects?

    - The discretization in 3. seems very relaxing: every numerical effect is replaced by the effect that the variable value is at least its lower bound. So basically any numerical variable can assume any value. This is never explicitly discussed.

    - The discretized action $\bar{a}$ is still a numeric action. But in 4. negative effects are removed, which makes only sense for actions with propositional variables. With 5. it becomes clear that apparently $(\geq ?x b_0)$ is a proposition. This is not clear beforehand.

    - The whole procedure of Algorithm 2 is basically a relaxed transformation from a numerical planning task to a classical planning task. What about other approaches that transform numerical planning tasks to a classical one? In the background it is said that Metric-FF can not deal with non-convex problems, but the problem definition does not say anything about that.

    - The heuristic module computes a plan for a transformed planning task (which is not clear when the mxPlanner approach is introduced). The transition module then takes the "original action $a_0$" from this sequence. What is an original action?

 - It is unclear why we require a transition module to compute state transitions.

 - It would be much easier to follow the gradient descent part if the paper would include informal descriptions of what exactly is happening. After all, the loss to satisfy numeric bounds penalizes any assignments to numeric variables that violate the variable bound.

 - Since obstacles are never formally introduced the loss to obstacles is not clear.

 - Given that the whole problem definition is not sound and complete, I don't see any of the sound and completeness properties of mxPlanner being guaranteed.

---

### Comment · Program_Chairs · 2020-09-14
**Final Decision: Reject**

Dear authors,

We are sorry to inform you that we reject your submission to the HSDIP workshop. The reviewers agree that the paper lacks in clarity, and also only partially fits the topic of the workshop. We hope that the reviews will allow you to work on an improved version that can be submitted to another venue.

Best,
The HSDIP'20 team

---

### Decision · Program_Chairs · 2020-09-30

Reject